# HuatuoGPT-II, One-stage Training for Medical Adaption of LLMs

**Junying Chen**[1,2], **Xidong Wang**[1,2], **Ke ji**[1,2], **Anningzhe Gao**[1*], **Feng Jiang**[2], **Shunian Chen**[1,2],
**Hongbo Zhang**[1,2] ,**Dingjie Song**[1,2], **Wenya Xie**[1,2], **Chuyi Kong**[1,2], **Jianquan Li**[2],
**Xiang Wan**[1,2], **Haizhou Li**[1,2], **Benyou Wang**[1,2*]
[1] Shenzhen Research Institute of Big Data
[2] The Chinese University of Hong Kong, Shenzhen
junying.chen.cs@gmail.com, gaoanningzhe@sribd.cn, wangbenyou@cuhk.edu.cn

## Abstract

Adapting a language model (LM) into a specific domain, *a.k.a* 'domain adaption',
is a common practice when specialized knowledge, e.g. medicine, is not encapsu-
lated in a general language model like Llama2. This typically involves a two-stage
process including *continued pre-training* and *supervised fine-tuning*. Implement-
ing a pipeline solution with these two stages not only introduces complexities
(necessitating dual meticulous tuning) but also leads to two occurrences of data dis-
tribution shifts, exacerbating catastrophic forgetting. To mitigate these, we propose
a one-stage domain adaption protocol where heterogeneous data from both the
traditional pre-training and supervised stages are unified into a simple instruction-
output pair format to achieve efficient knowledge injection. Subsequently, a data
priority sampling strategy is introduced to adaptively adjust data mixture during
training. Following this protocol, we train HuatuoGPT-II, a specialized LLM for
the medical domain in Chinese. HuatuoGPT-II achieve competitive performance
with GPT4 across multiple benchmarks, which especially shows the state-of-the-art
(SOTA) performance in multiple Chinese medical benchmarks and the newest phar-
macist licensure examinations. Furthermore, we explore the phenomenon of one-
stage protocols, and the experiments reflect that the simplicity of the proposed pro-
tocol improves training stability and domain generalization. Our code, data, and
models are available at https://github.com/FreedomIntelligence/HuatuoGPT-II.

## 1 Introduction

Currently, general large language model (LLM), such as the Llama series (Touvron et al., 2023), are
developing rapidly. Simultaneously, in some vertical domains[1], some researchers (Cui et al., 2023;
Yang et al., 2023c; Li et al., 2023a) attempt to develop specialized models. Specialized models have
the potential to yield results comparable to those of larger models by utilizing a medium-sized
model through the exclusion of certain general knowledge (Wang et al., 2024; Chen et al., 2023b;
Yang et al., 2023c). For instance, financial knowledge may not be sufficiently usefully in the medical
field and can be therefore omitted in moderately-sized medical LLMs, thereby freeing up more
capacity for memorizing medical knowledge.

---

[*]Benyou and Anningzhe are the corresponding authors.
[1] "Vertical domains" refer to specialized fields such as medicine, law, and finance, where training
focuses on deep, domain-specific knowledge and disregards other unnecessary domains. This contrasts with
horizontally trained models, which possess broader but shallower general knowledge.

**The two-stage protocol**    Adaption of general large language models in vertical domains typically involves two stages: **continued pre-training** and **supervised fine-tuning (SFT)** (Wang et al., 2023a). For this adaption in medicine domain, *continued pre-training* aims to inject specialized knowledge, such as medical expertise, while *supervised fine-tuning* seeks to activate the ability to follow medical instruction, as stated in Zhou et al., 2023.

**Issues of the two-stage protocol**    In the two-stage adaption, LLMs face two key challenge: 1) the difference in continual pre-training and supervised fine-tuning optimization objectives, and 2) the inherent difference in data distribution from general pre-training to continued pre-training. Therefore, the two-stage process suggests that the LLM experiences dual shifts in data distribution. As stated in Goodfellow et al., 2013, catastrophic forgetting occurs when neural networks learn multiple sequential tasks in a pipeline, resulting in the loss of knowledge from previously learned tasks. This problem worsens when the two-stage training involves significantly divergent data distributions and objectives (Bhat et al., 2022). Specifically, Cheng et al., 2023 argues continued pre-training on domain-specific corpora reduces the LLM's prompting capabilities. Secondly, the two-stage training pipeline adds complexity due to the interdependence of its stages. This intricacy not only complicates the optimization process but also limits scalability and adaptability. Each stage possesses distinct hyperparameters such as batch size, learning rate, and warmup procedures. These parameters necessitate careful manual selection through rigorous experimentation.

**The proposed one-stage protocol**    Following the philosophy of Parsimony, this work proposes a simpler protocol of domain adaption that unifies the two stages (continued pre-training and SFT) into a single stage. The core recipe is to transform domain-specific pre-training data into a unified format similar to fine-tuning data: a straightforward *(instruction, output)* pairs. This strategy diverges from the conventional dependence on unsupervised learning in continued pre-training, opting instead for a focused learning goal that emphasizes eliciting knowledge-driven responses to given instructions. The reformulated data is subsequently merged with fine-tuning data to facilitate one-stage domain adaption. In this process, we introduce a data priority sampling strategy aimed at initially learning domain knowledge from pre-training data and then progressively shifting focus to downstream fine-tuning data. This approach enhances the model's capability to utilize domain knowledge effectively.

**Verification for the new protocol**    To verify the one-stage protocol, we experiment on Chinese healthcare [2] where ChatGPT and GPT-4 perform relatively poorly (Wang et al., 2023a). Leveraging the proposed protocol, we train a Chinese medical language model, HuatuoGPT-II. Inspired by back-translation (Li et al., 2023d), we employ the prowess of LLMs for data unification, where all diverse and multilingual pre-training data are converted to Chinese instructions with a consistent style. This stage bridges the gap between two-stage data, especially for training LLMs in unpopular languages, where English data is overwhelmingly more abundant and high-quality. Subsequently, a priority sampling strategy is used to integrate pre-training and SFT instructions for domain adaption. We believe this unified domain adaptation protocol can be similarly effective in other specialized areas such as finance and law, as well as in different languages.

**Experimental Results**    Experimental results demonstrate that HuatuoGPT-II, a new Chinese medical language model, outperforms other open-source models and rivals proprietary ones like GPT-4 and ChatGPT in some Chinese medical Benchmarks. In expert manual evaluations, HuatuoGPT-II shows a remarkable win rate of 38% and tie in another 38% against GPT-4. Moreover, in a recent and untouched Chinese National Pharmacist Licensure Examination (2023), HuatuoGPT-II's superiority over other models, highlighting its specialized effectiveness in the medical field.

---

[2]This refers to using the Chinese language for healthcare, rather than Traditional Chinese Medicine.

Additionally, in a spectrum of evaluation methods, the one-stage protocol of HuatuoGPT-II prove more effective than the traditional two-stage training paradigm.

**Contributions**   The key contributions are:

- **A unified protocol for domain adaption**.  The paper introduces a simplified one-stage domain adaption protocol for training, streamlining the traditionally complex pipeline process.

- **A advanced Chinese medical LLM**. Our developed model, HuatuoGPT-II, leverages this protocol to achieve exceptional performance in Chinese healthcare domains, particularly in Traditional Chinese Medicine.

- **A novel generalization Test**. A novel benchmark using the fresh 2023 Chinese National Pharmacist Licensure Examination provides a robust assessment of HuatuoGPT-II, addressing test data leakage concerns and demonstrating superior performance of HuatuoGPT-II.

## 2   Data Collection

Domain data is typically divided into two parts: pre-training corpora and fine-tuning instructions.

**Medical Pre-training Corpus**   Domain corpus is pivotal for augmenting domain-specific expertise. We collect 1.1TB of both Chinese and English corpus, sourced from **encyclopedias**, **books**, **literature**, and **web corpus**, blending general corpora like C4 and specialized corpora such as PubMed.  All the corpora are publicly accessible, detailed in Appendix C. Then, a meticulous collection pipeline are established for curating high-quality domain data, detailed in the Appendix C. As a result, we obtained 5,252,894 premium medical documents for the pre-training corpus.

**Medical Fine-Tuning Instructions**   For the fine-tuning instruction, We acquire 142K real-world medical questions as instructions from Huatuo-26M (Li et al., 2023c), and had GPT-4 respond to them as outputs.  The fine-tuning instruction is utilized to generalize the model's capability to interact with users within the domain.

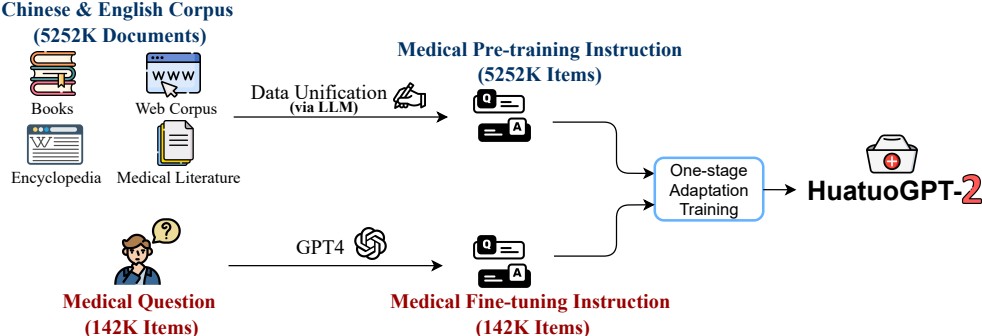

Figure 1: Schematic of One-stage Adaption of HuatuoGPT-II.

## 3   One-stage Adaption

One-stage Adaption strategy aims to unify the conventional Two-stage Adaption process (continued pre-training and supervised fine-tuning) into a single stage, as shown in Figure 1. The adaption process can be executed in two succinct steps: 1) Data unification and 2) One-stage training.

Subsequently, we will detail our method for adapting to the Chinese healthcare sector and developing and the development of HuatuoGPT-II.

### 3.1   Data Unification

Domain pre-training corpus is pivotal for augmenting domain-specific expertise. However, it faces challenges such as diverse languages and genres, punctuation errors, and ethical concerns in its pre-training corpus. Particularly, there's a noticeable discrepancy between its unsupervised training and the supervised instruction learning in Supervised Fine-Tuning (SFT). Data Unification aims to unify this data into a consistent format, aligning it with SFT data. Traditional methods like those in Cheng et al., 2023 fall short due to the variation in language and genre. Hence, we leverage Large Language Models to achieve effective data unification.

Our method of data unification is straightforward yet effective. We generate instructions based on the text of the corpus, and then we generate responses based on both the corpus and the instructions. The prompt for generating instructions is shown in Figure 2.

> Please create a <question> that closely aligns with the provided <text>. Ensure that the <question> is formulated in  [target language]
>
> and does not explicitly reference the text. You may incorporate specific scenarios or contexts in the <question>, allowing the <text> to serve as a comprehensive and precise answer.
>
> <text>:  [domain-specific corpus]
>
> <question>:

Figure 2:   The  prompt  for  question  generation.   [target language]  is  Chinese,  and  [domain-specific corpus]  refers to a corpus in the domain-specific pre-training corpora.

After obtaining questions from the corpus text, we use the prompt, shown in Figrue 3, to let a Large Language Model (LLM) generate responses based on the questions and the corpus. Therefore, all pre-training corpora are converted into a instruction-output format, identical to our single-turn SFT data. An example of our final SFT data is shown in Table 7.

Here, we use ChatGPT as the LLM for data unification, converting all the medical corpus into instructions of the same language and genre. As detailed in Appendix F, Data Unification can be achieved independently of external LLMs. In Appendix E, we verified that the selection of LLM for data unification is inconsequential. This strategy also mitigates potential ethical concerns inherent in the corpus. Additionally, we also use statistical and semantic recognition to keep the model-generated answers close to the data's original content, as explained in Appendix H.

### 3.2   One-stage Training

In the one-stage training process, we integrate data from the Medical Pre-training Instruction and Medical Fine-tuning Instruction datasets to form dataset $D$. The traditional two-stage pipeline adopts a completely separate form, first learning knowledge and then learning instructions to

You are [model name], equipped with in-depth knowledge in [domain]. Your task is to directly answer the user's <question> in [target language]. In formulating your response, you must thoughtfully reference the <reference text>, ensuring that your reply does not disclose your reliance on <reference text>. Aim to provide a comprehensive and informative response, incorporating relevant insights from <reference text> to best assist the user. Please be cautious to avoid including any content that might raise ethical concerns.

<question>: [question generated by LLM]

<reference text>: [domain-specific corpus]

<reply>:

Figure 3: Prompt for answer generation. [model name] refers to HuatuoGPT-II, [domain] is medicine, and [question generated by LLM]] is the previously text-derived query.

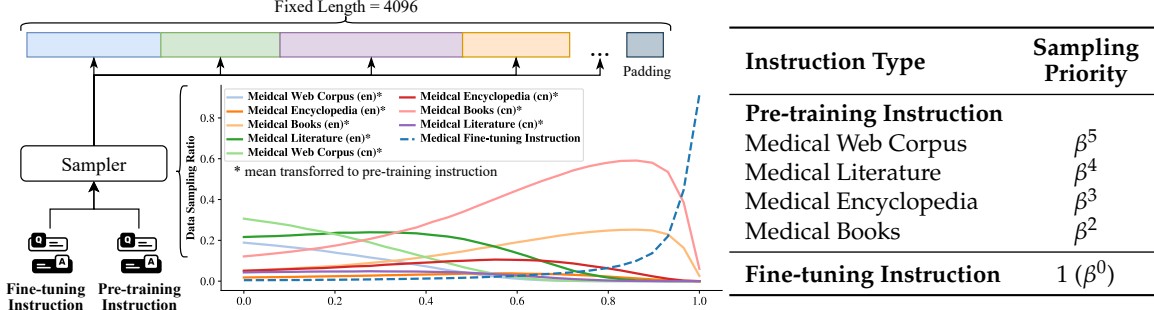

Figure 4: One-stage training process and sampling priority. The diagram (left) illustrates the one-stage training methodology. The table (right) details the sampling priority for various instructional data types, with the $\beta$ (set as 2) dictating the relative priority. Higher $\beta$ values indicate a more sequential sampling approach, whereas lower values suggest a blended strategy.

follow. It exacerbates the catastrophic forgetting of the knowledge acquired in the first stage (Bhat et al., 2022). Referring to Touvron et al., 2023, simply mixing all data could hinder the ability of Large Language Models (LLMs). To address this, we introduce a priority sampling strategy in this study. This strategy starts with domain knowledge, gradually transitioning to fine-tuning data while reducing the sampling ratio of previously learned data over time.

In the priority sampling strategy, the sampling probability of each data $x \in D$ changes over the course of training. The sampling probability of data $x$ at step $t$ during training is:

$$P_t(x) = \frac{\pi(x)}{\sum_{y \in D - S_t} \pi(y)}$$

Here, $\pi(x)$ denotes the priority of element $x$, and $S_t$ represents the sampled data before step $t$. The priority setting and data sampling distribution are delineated in Figure 4. Notably, the priority $\pi(x)$ is static, whereas the sampling probabilities of each data source dynamically changes. More precisely, consider data sources $D_1^t, D_2^t, \ldots, D_n^t \subseteq D - S^t$ at time $t$, each with an assigned priority

$\pi(x \in D_i^t) = \beta^{K_i}$. The probability of selecting an element from $D_i^t$ is given by:

$$P_t(x \in D_i^t) = \frac{|D_i^t|\beta^{K_i}}{\sum_{j=1}^{n} |D_j^t|\beta^{K_j}}$$

After an element is selected from $D_i^t$, the size of $D_i^{t+1}$ becomes $|D_i^{t+1}| = |D_i^t| - 1$, resulting in: $P_{t+1}(x \in D_i^{t+1}) < P_t(x \in D_i^t)$. Therefore, each selection from $D_i$ slightly decreases its sampling probability, leading to a dynamic update. The parameter $\beta$ plays a crucial role in adjusting the sampling intensity among high-priority sources, with higher $\beta$ values favoring sequential sampling, while lower values encourage mixed sampling.

To enable the model to first learn domain capabilities and then gradually shift to instruction interaction learning, we assign a higher priority to pre-training instruction data. Furthermore, to facilitate the model's transition from low to high knowledge density learning, we assign different priorities to the four types of data sources. Appendix I shows details of how we determine sampling priority values for different types of data.

## 4 Experiments

### 4.1 Experimental settings

**Base Model & Setup**   HuatuoGPT-II, tailored for Chinese healthcare, builds upon the foundations o f the **Baichuan2-7B-Base** and **Baichuan2-13B-Base** models. Since all data consists solely of single-round format instruction, we enrich the Medical Fine-tuning Instruction dataset by integrating ShareGPT data [3]. This allows HuatuoGPT-II to support multi-round dialogues while maintaining its general domain capabilities. For training details, please see the Appendix J.

**Baselines**   We compare HuatuoGPT-II with several leading general large language models known for their excellent general chat capabilities in Chinese. These models are **Baichuan2-7B/13B-Chat**(Yang et al., 2023a), **Qwen-7B/14B-Chat**(Bai et al., 2023) and **ChatGLM3-6B**(Zeng et al., 2023). Additionally, for the Chinese medical context, we carefully select **DISC-MedLLM**(Bao et al., 2023) and **HuatuoGPT**(Zhang et al., 2023) based on an experimental experiment detailed in Appendix R. We also consider leading proprietary models, including **ERNIE Bot** ( 文心一言) (Sun et al., 2021), **ChatGPT**(OpenAI, 2022), and **GPT-4**[4](OpenAI, 2023), noted for their extensive parameters and superior performance. For the details of these models, please refer to Appendix M.

### 4.2 Medical Benchmark

In this section, we evaluate the medical capabilities of HuatuoGPT-II on popular benchmarks. We focus on four medical benchmarks (MedQA, MedMCQA, CMB, CMExam) and two general benchmarks (C-Eval, CMMLU), focusing specifically on their medical components. See Appendix K for more details on these benchmarks. To ensure fairness, we use the most straightforward prompt for all the models, as shown in Appendix K.

**Medical Benchmark**   As shown in Table 1, the benchmark results highlight HuatuoGPT-II's impressive proficiency in the medical domain. Its exceptional performance in Chinese medical benchmarks

---

[3] https://huggingface.co/datasets/philschmid/sharegpt-raw
[4] The versions are gpt-3.5-turbo-0613 and gpt-4-0613. The same version we use in data unification.

| Model | English | | Chinese | | | | Average |
| --- | --- | --- | --- | --- | --- | --- | --- |
| | MedQA | MedMCQA | CMB | CMExam | CMMLU$_{Med.}$ | C_Eval$_{Med.}$ | |
| DISC-MedLLM | 28.67 | - | 32.47 | 36.62 | - | - | - |
| HuatuoGPT | 25.77 | 31.20 | 28.81 | 31.07 | 33.23 | 36.53 | 31.92 |
| ChatGLM3-6B | 28.75 | 35.91 | 39.81 | 43.21 | 46.97 | 48.80 | 40.73 |
| Baichuan2-7B-Chat | 33.31 | 38.90 | 46.33 | 50.48 | 50.74 | 51.47 | 45.39 |
| Baichuan2-13B-Chat | 39.43 | 41.86 | 50.87 | 54.90 | 52.95 | 58.67 | 49.77 |
| Qwen-7B-Chat | 33.46 | 41.36 | 49.39 | 53.33 | 54.65 | 52.80 | 47.50 |
| Qwen-14B-Chat | 42.81 | 46.59 | 60.28 | 63.57 | **64.55** | **65.07** | 57.16 |
| ChatGPT (API) | **52.24** | **53.60** | 43.26 | 46.51 | 50.37 | 48.80 | 48.51 |
| HuatuoGPT-II (7B) | 41.13 | 41.87 | 60.39 | 65.81 | 59.08 | 62.40 | 55.13 |
| HuatuoGPT-II (13B) | 45.68 | 47.41 | **63.34** | **68.98** | 61.45 | 64.00 | **58.47** |

Table 1: Medical benchmark results. *Med.* signifies extraction of only medical-related questions. '-' indicate that the model cannot follow the question and make a choice. Due to the too large size of these benchmarks, we exclude the testing of GPT-4 and ERNIE Bot here.

like CMB and CMExam reflects its deep understanding of medical concepts in a Chinese context. Moreover, HuatuoGPT-II maintains a high level on English benchmarks, demonstrating its medical capabilities. To further evaluate the medical expertise, we utilize a dataset with various Chinese national medical exams. The results, displayed in Table 4, reveal that HuatuoGPT-II surpass all open-source models and closely rivaled the proprietary model, ERNIE Bot. In Appendix B, we present the performance of HuatuoGPT-II on additional medical benchmarks.

| Model | 2023 Pharmacist Licensure Examination (Pharmacy) | | | | | 2023 Pharmacist Licensure Examination (TCM) | | | | | Avg. |
| --- | --- | --- | --- | --- | --- | --- | --- | --- | --- | --- | --- |
| | Optimal Choice | Matched Selection | Integrated Analysis | Multiple Choice | Total Score | Optimal Choice | Matched Selection | Integrated Analysis | Multiple Choice | Total Score | |
| DISC-MedLLM | 22.2 | 26.8 | 23.3 | 0.0 | 22.6 | 24.4 | 32.3 | 15.0 | 0.0 | 24.9 | 23.8 |
| HuatuoGPT | 25.6 | 25.5 | 23.3 | 2.6 | 23.4 | 24.1 | 26.8 | 31.6 | 7.5 | 24.9 | 24.2 |
| ChatGLM3-6B | 39.5 | 39.1 | 10.5 | 0.2 | 34.6 | 31.8 | 38.2 | 25.0 | 20.0 | 32.9 | 33.8 |
| Qwen-7B-chat | 43.8 | 46.8 | 33.3 | 18.4 | 41.9 | 40.0 | 43.2 | 33.3 | 17.5 | 38.8 | 40.4 |
| Qwen-14B-chat | 56.2 | 58.6 | 41.7 | 21.1 | 52.7 | 51.3 | 51.0 | 27.5 | 41.7 | 47.9 | 50.3 |
| Baichuan2-7B-Chat | 51.2 | 50.9 | 30.0 | 2.6 | 44.6 | 48.1 | 46.0 | 35.0 | 7.5 | 42.1 | 43.4 |
| Baichuan2-13B-Chat | 43.8 | 52.7 | 36.7 | 7.9 | 44.2 | 41.3 | 46.4 | 43.3 | 15.0 | 41.7 | 43.0 |
| ERNIE Bot (API) | 45.0 | 60.9 | 36.7 | 23.7 | 49.6 | 53.8 | 59.1 | 38.3 | 20.0 | 51.5 | 50.6 |
| ChatGPT (API) | 45.6 | 44.1 | 36.7 | 13.2 | 41.2 | 34.4 | 32.3 | 30.0 | 15.0 | 31.2 | 36.2 |
| GPT-4 (API) | 65.1 | 59.6 | 46.7 | 15.8 | **57.3** | 40.6 | 42.7 | 33.3 | 17.5 | 38.8 | 48.1 |
| HuatuoGPT-II(7B) | 41.9 | 61.0 | 35.0 | 15.7 | 47.7 | 52.5 | 51.4 | 41.7 | 15.0 | 47.5 | 47.6 |
| HuatuoGPT-II(13B) | 47.5 | 64.1 | 45.0 | 23.7 | 52.9 | 48.8 | 61.8 | 45.0 | 17.5 | **51.6** | **52.3** |

Table 2: Results of the 2023 Chinese National Pharmacist Licensure Examination. It consists of two tracks of Pharmacy track and Traditional Chinese Medicine (TCM) Pharmacy track.

**The Fresh Medical Exams** Tackling data contamination in LLM training is challenging (Huang et al., 2023), especially with extensive and intricate training data. To counter this, we selected the 2023 Chinese National Pharmacist Licensure Examination, held on October 21, 2023, as our benchmarks. This date is crucially before both our data collection cut-off (October 7, 2023) and the release of our assessment models. The annual uniqueness of the exam questions ensures a reliable safeguard against contamination risks. The results in Table 2 show that HuatuoGPT-II ranked second after GPT-4 in the Pharmacy track. However, in the Traditional Chinese Medicine track, HuatuoGPT-II led with 51.6 points. HuatuoGPT-II demonstrated superior performance in average scores across both tracks, highlighting its proficiency in the medical field.

### 4.3 Medical Response Quality

To evaluate the model's performance in real-world medical scenarios, we utilize real-wrold questions in both single-round and multi-round formats, sourced respectively from KUAKE-QIC (Zhang et al., 2021) and Med-dialog (Zeng et al., 2020), following the approach of HuatuoGPT (Zhang et al., 2023). The assessment details are provided in the Appendix L.

| HuatuoGPT-II(7B) vs Other Model | Single-round QA | | | Multi-round Dialogue | | | Average Win/Tie Rate |
|---|---|---|---|---|---|---|---|
| | **Win** | **Tie** | **Fail** | **Win** | **Tie** | **Fail** | |
| **Automated Evaluation Using GPT-4** | | | | | | | |
| HuatuoGPT-II(7B) vs GPT-4 | **58** | 21 | 21 | **62** | 15 | 23 | 78.0% |
| HuatuoGPT-II(7B) vs ChatGPT | **62** | 18 | 20 | **69** | 14 | 17 | 81.5% |
| HuatuoGPT-II(7B) vs Baichuan2-13B-Chat | **64** | 14 | 22 | **75** | 11 | 14 | 82.0% |
| HuatuoGPT-II(7B) vs HuatuoGPT | **87** | 7 | 6 | **67** | 15 | 18 | 88.0% |
| **Human Expert Evaluation** | | | | | | | |
| HuatuoGPT-II(7B) vs GPT-4 | **38** | 38 | 24 | **53** | 17 | 30 | 73% |
| HuatuoGPT-II(7B) vs ChatGPT | **52** | 33 | 15 | **56** | 11 | 33 | 76% |
| HuatuoGPT-II(7B) vs Baichuan2-13B-Chat | **63** | 19 | 18 | **63** | 19 | 18 | 82% |
| HuatuoGPT-II(7B) vs HuatuoGPT | **81** | 11 | 8 | **68** | 6 | 26 | 83% |

Table 3: Results of the Automated Evaluation Using GPT-4 and Expert Evaluation.

**Automatic Evaluation**    We utilize GPT-4 to evaluate which of the two models generated better outputs. The results, as indicated in Table 3 (for more compared baselines, see Table 13), show that HuatuoGPT-II has a higher win rate compared to other models. Notably, HuatuoGPT-II achieve a higher win rate in comparison with GPT-4. Although its fine-tuning data originated from GPT-4, its extensive medical corpus provided it with more medical knowledge, as shown in Table 2.

**Expert Evaluation**    For further evaluating the quality of the model outputs, we invite four licensed physicians to score them, with the detailed criteria available in the Appendix L.2. Due to the high cost of expert evaluation, we select four models for comparison with HuatuoGPT-II. Results, as outlined in Tables 3, indicate that HuatuoGPT-II consistently outperforms its peers, aligning with automatic evaluation. This consensus between expert opinions and GPT-4's evaluations underscores HuatuoGPT-II's efficacy in medical response generation. The case study on the model's response can be found in Appendix S.

### 4.4 One-stage Vs. Two-stage Adaption

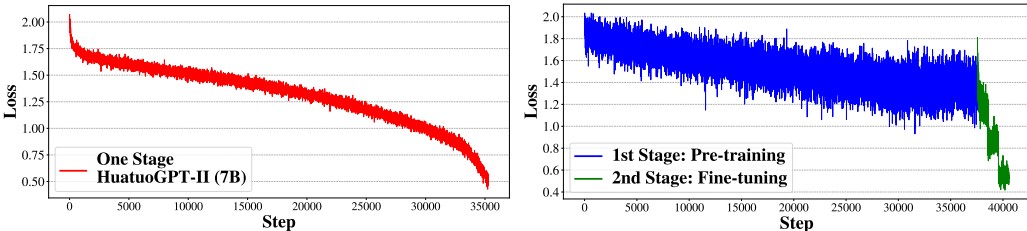

Figure 5: Comparison of the loss outcomes between proposed One-stage Adaption and conventional Two-stage Adaption using the same SFT data and medical corpus.

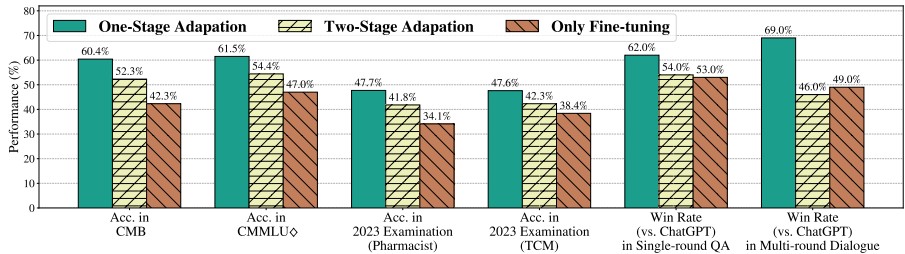

Figure 6: The comparison results of One-stage Adaption and Two-stage Adaption. "Only Fine-tuning" refers to the model that only fine-tunes the backbone directly. Win Rate is the result scored by automatic evaluation using GPT-4.

To validate the superiority of One-stage Adaption, we conduct a traditional Two-stage Adaption to fine-tune the the Baichuan2-7B-Base on the same medical data.

Figure 5 shows the training loss of these two training methods. The loss of Two-stage Adaption shows instability, marked by pronounced fluctuations and loss spikes. This instability arises from the varied content and styles within the medical pre-training corpus, which includes four distinct data types as detailed in Table 9. The variation is further amplified by the differences between Chinese and English datasets. Our one-stage Adaption unifies the diverse contents and styles of the pre-training corpus, improving training stability and reducing loss fluctuation. Moreover, due to data discrepancies between the two stages, there is a noticeable loss divergence between the pre-training and fine-tuning stages in Two-stage Adaption. In contrast, One-stage Adaption effectively resolves this issue by simplifying the pre-training corpus into a unified language and style, aligning with SFT data, making a more stable and smooth training process.

The results of the previously mentioned experiments also demonstrate that One-stage Adaption achieves better performance than other methods, as shown in Figure 6. It can be seen that on all six datasets, our One-stage Adaption performance is significantly better than the Two-stage Adaption performance (from 5.3% to 23%), especially in single-round Q&A and multi-round conversation tasks. This superiority is likely due to two-stage unification and more effective knowledge generalization in One-stage Adaption.

### 4.5 The Relative Priority for Sampling

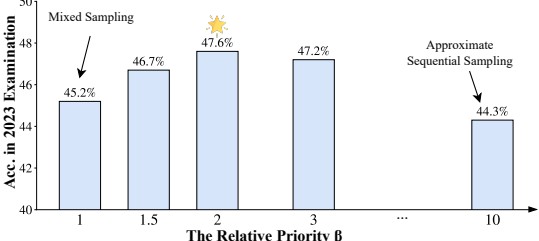

Figure 7: Comparison of model performance under different relative priority $\beta$ settings.

In evaluating the priority sampling strategy, we tested it across different $\beta$ settings using the same HuatuoGPT-II (7B) configuration. Our results show that model performance drastically drops when $\beta = 0$ (mixed sampling) or when $\beta$ is excessively high (Sequential Sampling), highlighting the importance of priority sampling. The best performance is achieved when $\beta$ is around 2.

Moreover, we also conducted an ablation study on data unification in appendix G. The experiment indicates that data unification is crucial for the performance improvement of the one-stage training.

## 5   Conclusion

In this work, we propose a one-stage domain adaption method, simplifying the conventional two-stage domain adaption process and mitigating its associated challenges. This approach is straightforward, involving the use of LLM capabilities to align domain corpus with SFT data and adopting a priority sampling strategy to enhance domain adaption. Based on this method, we develop a Chinese medical language model, HuatuoGPT-II. In the experiment, HuatuoGPT-II demonstrates state-of-the-art performance in the Chinese medicine domain across various benchmarks. It even surpasses proprietary models like ChatGPT and GPT-4 in some aspects, particularly in Traditional Chinese Medicine. The experimental results also confirm the reliability of the one-stage domain adaption method, which shows a significant improvement over the traditional two-stage performance. This One-stage Adaption promises to offer valuable insights and a methodological framework for future LLM domain adaption works.

## Ethic Statement

The development of HuatuoGPT-II, a specialized language model for Chinese healthcare, raises several potential risks.

**Accuracy of Medical Advice**   While HuatuoGPT-II has shown promising results in the domain of Chinese healthcare, it's crucial to underscore that at this stage, it should not be used to provide any medical advice. This caution stems from the inherent limitations of large language models, including their capacity for generating plausible yet inaccurate or misleading information.

**Data Privacy and Ethics**   The ethical handling of data is paramount, especially in the sensitive field of healthcare. The primary data sources for HuatuoGPT-II include medical texts, such as textbooks and literature, ensuring that patient-specific data is not utilized. This approach aligns with ethical guidelines and privacy regulations, ensuring that individual patient information is not compromised. Another significant aspect of our methodology is the 'data unification' process, which aims to address potential ethical issues in the training data. By employing large language models to rewrite the medical corpora, we aim to eliminate any ethically questionable content, thereby ensuring that the training process and the model align with ethical standards.

## Acknowledgement

This work was supported by the Shenzhen Science and Technology Program (JCYJ20220818103001002), Shenzhen Doctoral Startup Funding (RCBS20221008093330065), Tianyuan Fund for Mathematics of National Natural Science Foundation of China (NSFC) (12326608), Shenzhen Key Laboratory of Cross-Modal Cognitive Computing (grant number

ZDSYS20230626091302006), and Shenzhen Stability Science Program 2023, Shenzhen Key Lab of Multi-Modal Cognitive Computing.

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

# A Related Work

## A.1 Domain adaption

Recent research has also indicated that such domain adaption using further training Cheng et al., 2023 leads to a drastic drop despite still benefiting fine-tuning evaluation and knowledge probing tests. This inspires us to design a different protocol for domain adaption. Also, Gunasekar et al. (2023); Li et al. (2023e); Chen et al. (2023c); Kong et al. (2023) show a possibility that the 10B well-selected dataset could achieve comparable performance to a much larger model.

## A.2 Medical LLMs

The rapid advancement of large language models (LLMs) in Chinese medical field is driven by the release of open-source Chinese LLMs, notably trained via instruction fine-tuning. Doctor-GLM (Xiong et al., 2023) and MedicalGPT (Xu, 2023) are fine-tuned on various Chinese and English medical dialogue datasets. Another Chinese medical LLM, BenTsao (Wang et al., 2023b) is fine-tuned on distilled data derived from knowledge graphs. Bianque-2 (Chen et al., 2023a) includes multiple rounds of medical expansions, encompassing drug instructions and encyclopedia knowledge instructions. ChatMed-Consult (Zhu & Wang, 2023) is fine-tuned on both Chinese online consultation data and ChatGPT responses. DISC-MedLLM (Bao et al., 2023) is fine-tuned on more than 470,000 medical data including doctor-patient dialogues and knowledge QA pairs.

By integrating the reinforcement learning, HuatuoGPT (Zhang et al., 2023) is fine-tuned on a 220,000 medical dataset consisting of ChatGPT-distilled instructions and dialogues alongside real doctor-patient single-turn Q&As and multi-turn dialogues. ZhongJing (Yang et al., 2023b) undergoes a comprehensive three-stage training process: continuous pre-training on various medical data, instruction fine-tuning on single-turn and multi-turn dialogue data as well as medical NLP tasks data, and reinforcement learning adjudicated by experts to ensure reliability and safety.

# B Supplementary Experiment

| | Traditional Chinese Medicine | | | | | | | | | Clinical | | | | | | |
| Model | Assistant | | | Physician | | | Pharmacist | | | Assistant | | | Physician | | | Avg. |
| | 2015 (300) | 2016 (220) | 2017 (116) | 2012 (600) | 2013 (600) | 2016 (430) | 2017 (168) | 2018 (167) | 2019 (223) | 2018 (234) | 2019 (244) | 2020 (244) | 2018 (449) | 2019 (476) | 2020 (436) | |
|---|---|---|---|---|---|---|---|---|---|---|---|---|---|---|---|---|
| HuatuoGPT | 26.0 | 30.9 | 32.8 | 31.3 | 26.8 | 30.2 | 18.4 | 27.5 | 25.1 | 32.5 | 33.6 | 27.5 | 32.7 | 28.6 | 30.1 | 28.9 |
| DISC-MedLLM | 38.3 | 41.8 | 26.7 | 36.2 | 38.7 | 35.1 | 25.0 | 22.2 | 22.0 | 49.2 | 36.1 | 41.8 | 41.4 | 36.6 | 35.8 | 35.1 |
| ChatGLM3-6B | 45.7 | 45.0 | 50.0 | 46.3 | 45.8 | 46.5 | 42.3 | 28.1 | 35.4 | 50.9 | 48.8 | 43.9 | 41.7 | 43.9 | 43.6 | 43.9 |
| Baichuan2-7B-Chat | 57.3 | 57.3 | 58.6 | 55.7 | 58.5 | 57.9 | 41.7 | 41.9 | 45.7 | 61.1 | 55.7 | 55.3 | 51.0 | 53.6 | 50.0 | 53.4 |
| Baichuan2-13B-Chat | 64.7 | 58.2 | 62.9 | 61.7 | 61.5 | 63.3 | 54.2 | 38.9 | 48.4 | 66.2 | 64.8 | 63.1 | 65.9 | 58.8 | 61.5 | 59.6 |
| Qwen-7B-Chat | 54.7 | 55.9 | 56.0 | 52.7 | 53.5 | 54.4 | 44.0 | 33.5 | 43.0 | 68.8 | 63.9 | 57.8 | 60.6 | 57.6 | 54.1 | 54.0 |
| Qwen-14B-Chat | 65.3 | 63.2 | 67.2 | 64.8 | 63.3 | 67.9 | 54.8 | 49.1 | 52.5 | 74.8 | 75.0 | 69.3 | 73.7 | 69.7 | 68.8 | 65.3 |
| ERNIE Bot (API) | 73.3 | 66.3 | 73.3 | 70.0 | 71.8 | 66.7 | 55.9 | 50.3 | 60.0 | 78.2 | 77.0 | 77.5 | 66.6 | 70.8 | 74.1 | 68.8 |
| ChatGPT (API) | 46.0 | 36.4 | 41.4 | 36.7 | 38.5 | 40.5 | 32.1 | 28.1 | 30.0 | 63.3 | 57.8 | 53.7 | 53.7 | 52.5 | 51.8 | 44.2 |
| GPT-4 (API) | 47.3 | 48.2 | 53.5 | 50.3 | 53.7 | 54.2 | 41.1 | 43.7 | 48.0 | 79.9 | 72.5 | 70.9 | 74.8 | 73.1 | 68.4 | 58.6 |
| **HuatuoGPT-II (7B)** | 67.1 | 65.2 | 67.5 | 67.9 | 67.4 | 64.9 | 53.0 | 46.7 | 51.0 | 70.9 | 73.2 | 69.4 | 68.8 | 66.5 | 67.7 | 64.5 |
| **HuatuoGPT-II (13B)** | 70.3 | 70.0 | 71.6 | 71.0 | 69.2 | 70.2 | 56.5 | 52.1 | 54.7 | 73.1 | 76.6 | 70.1 | 72.8 | 68.9 | 72.2 | 68.0 |

Table 4: The results of Chinese National Medical Licensing Examinations. The year represents the actual examination year. Note that the exam here may not be complete, and the blue fonts indicates the number of questions.

**Chinese Medical Licensing Examination** To evaluate HuatuoGPT-II's medical competence, we utilize a dataset comprising medical examination questions from China and the United States. This approach aims to measure the model's adaptability and proficiency across varied medical knowledge frameworks and terminologies. The results of the Chinese National Medical Licensing Examination are outlined in Table 4. In the results, HuatuoGPT-II (13B) not only surpassed all open-source models but also closely approached the performance of the leading proprietary model, ERNIE Bot. This notable outcome reflects the model's advanced understanding of Chinese medical principles and its adeptness in applying this knowledge in complex examination contexts. The 13B variant's elevated scores signify enhanced analytical capabilities and deeper comprehension of the nuances inherent in Chinese medical practice.

**CMB-Clin** CMB-Clin is a dataset designed to evaluate the Clinical Diagnostic capabilities of LLMs, based on 74 classical complex and real-world cases derived from textbooks. Distinct from the response quality evaluations, CMB-Clin provides standard answers for reference and scores each model individually. We follow the same evaluation strategy from the original paper which utilizes GPT-4 as the evaluator. Results, as delineated in Table 5, indicate that HuatuoGPT-II outperforms its counterparts, excluding GPT-4. Intriguingly, the 7B version of HuatuoGPT-II demonstrates superior efficacy over its 13B variant, a phenomenon potentially attributable to foundational capacity variances, as evidenced by Baichuan2-7B-Chat's superior performance compared to Baichuan2-13B-Chat.

| Model | Fluency | Relevance | Completeness | Proficiency | Avg.↑ |
|---|---|---|---|---|---|
| GPT-4 | 4.95 | 4.71 | 4.35 | 4.66 | 4.67 |
| **HuatuoGPT-II (7B)** | 4.94 | 4.56 | 4.24 | 4.46 | 4.55 |
| ERNIE Bot | 4.92 | 4.53 | 4.16 | 4.55 | 4.54 |
| ChatGPT | 4.97 | 4.49 | 4.12 | 4.53 | 4.53 |
| **HuatuoGPT-II (13B)** | 4.92 | 4.38 | 4.00 | 4.40 | 4.43 |
| Baichuan2-7B-Chat | 4.93 | 4.41 | 4.03 | 4.36 | 4.43 |
| Qwen-14B-Chat | 4.90 | 4.35 | 3.93 | 4.48 | 4.41 |
| Qwen-7B-Chat | 4.94 | 4.17 | 3.67 | 4.33 | 4.28 |
| Baichuan2-13B-Chat | 4.88 | 4.18 | 3.78 | 4.27 | 4.28 |
| ChatGLM3-6B | 4.92 | 4.11 | 3.74 | 4.23 | 4.25 |
| HuatuoGPT | 4.89 | 3.76 | 3.38 | 3.86 | 3.97 |
| DISC-MedLLM | 4.82 | 3.24 | 2.75 | 3.51 | 3.58 |

Table 5: Results of CMB-Clin on Automatic Evaluation using GPT-4.

**USMLE** The United States Medical Licensing Examination (USMLE) outcomes, as delineated in Table 6, reveal that HuatuoGPT-II (13B) outperforms comparative open-source models. Despite a discernible disparity with ChatGPT, it is important to note that the USMLE's English-centric nature imposes constraints on HuatuoGPT-II, primarily designed for Chinese medical contexts. However, its commendable performance in the USMLE underscores its proficiency in employing medical knowledge across diverse scenarios, effectively addressing the range of challenges posed by the USMLE.

## C Domain Corpus Collection

Domain corpus is pivotal for augmenting domain-specific expertise. Domain data necessitates a comprehensive collection of high-quality and domain-specific content, surpassing the scope of the

| Models | Stage1 (6308) | Stage2&3 (5148) | Avg. (11456) |
|---|---|---|---|
| HuatuoGPT | 28.68 | 28.38 | 28.54 |
| ChatGLM3-6B | 33.39 | 32.49 | 32.98 |
| Baichuan2-7B-Chat | 38.11 | 37.32 | 37.76 |
| Baichuan2-13B-Chat | 44.99 | 45.57 | 45.25 |
| Qwen-7B-Chat | 40.90 | 36.09 | 38.73 |
| Qwen-14B-Chat | 48.73 | 42.45 | 45.90 |
| ChatGPT (API) | **57.04** | **56.27** | **56.69** |
| **HuatuoGPT-II(7B)** | 45.72 | 44.45 | 45.15 |
| **HuatuoGPT-II(13B)** | 47.34 | 49.37 | 48.25 |

Table 6: The results of The United States Medical Licensing Examination (USMLE) from MedQA. The blue fonts indicate the number of questions.

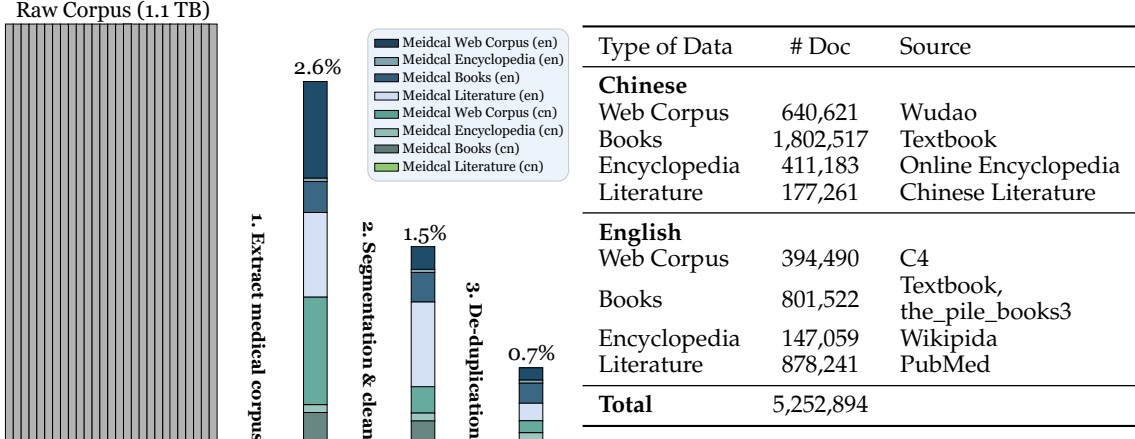

Figure 8: Domain Data collection.

Figure 9: Summary of the Medical Pre-training Corpus.

| Type of Data | # Doc | Source |
|---|---|---|
| **Chinese** | | |
| Web Corpus | 640,621 | Wudao |
| Books | 1,802,517 | Textbook |
| Encyclopedia | 411,183 | Online Encyclopedia |
| Literature | 177,261 | Chinese Literature |
| **English** | | |
| Web Corpus | 394,490 | C4 |
| Books | 801,522 | Textbook, the_pile_books3 |
| Encyclopedia | 147,059 | Wikipida |
| Literature | 878,241 | PubMed |
| **Total** | 5,252,894 | |

foundational corpus. We identified four primary data types for extensive collection: encyclopedias, books, academic literature, and web content. Focusing on these categories, we amassed 1.1TB of Chinese and English data, as shown in Table 9.

The domain data collection pipeline is an essential part of ensuring the quality of domain language corpora, designed to extract high-quality and diverse domain corpora from large-scale corpora. The methodology encompasses four primary steps:

1. **Extract Medical Corpus:** This process aims to remove irrelevant domain corpora, serving as the first step in filtering massive corpora. We employ a dictionary-based approach, obtaining dictionaries containing medical vocabulary from THUOCL[5] and The SPECIALIST Lexicon[6] from the Unified Medical Language System. We strive to exclude non-medical

---

[5]https://github.com/thunlp/THUOCL
[6]https://www.nlm.nih.gov/research/umls/new_users /online_learning/LEX_001.html

terms to form a domain-specific dictionary. For each text segment, we evaluate whether it's domain-specific by assessing the density of matched domain words from the dictionary. This dictionary method is an effective and efficient way to extract domain text.

2. **Segmentation:** Since we need to convert the corpora into instructions and do the data cleaning and de-duplication, it's necessary to segment all corpora into fragments. We divide each text into sentences, then use a moving window to turn the corpus into segments with a length limit, ensuring no loss of information by including sentences before and after the window.

3. **Cleaning:** We notice a significant proportion of medical corpora is related to medical advertising, despite appearing fluent, contains little domain knowledge and introduces bias. To filter out medical advertisement texts and low-quality texts, we utilize ChatGPT to select problematic texts and train a corpus quality classification model to clean corpora.

4. **De-duplication:** De-duplication is a crucial step in corpus processing, as domain knowledge often has significant redundancy. We use a sentence embedding model to convert corpora into embeddings and then employ dense retrieval methods to remove semantically similar texts.

Our data sources consist of four categories: (1) Web Corpus, which includes C4 (Raffel et al., 2020) and Wudao (Yuan et al., 2021); (2) Books, primarily comprising Textbook and the_pile_books3(Gao et al., 2020); (3) Encyclopedia, encompassing Chinese Medical Encyclopedia and Wikipedia; (4) Medical Literature, which consists of PubMed and Chinese literature. Following the aforementioned steps, the corpus is converted into 5,252K of instruction data.

## D  Example of Data Unification

Table 7 illustrates the process of generating instruction and response based on documents in Data Unification.

| Description | content |
| --- | --- |
| Raw Data (from English Textbook) | It is one of the most frequent mental illnesses, making up a large proportion of persons in assisted living and skilled nursing facilities. Although Alzheimer disease has been described at every period of adult life, the majority of patients are in their sixties or older; a relatively small number have been in their late fifties or younger.... |
| Query of Data Unification | 什么是阿尔茨海默病的主要临床和病理特征？( Translation: What are the main clinical and pathological features of Alzheimer's disease?) |
| Answer of Data Unification | 阿尔茨海默病是一种最常见的精神疾病，占据了较大比例的辅助生活和专业护理设施的人群。虽然阿尔茨海默病在成年期的各个阶段都可能出现，但大多数患者在六十岁或以上；只有一小部分在五十多岁甚至更年轻...(Translation: Alzheimer's disease is one of the most common mental illnesses, occupying a larger percentage of the population in assisted living and skilled nursing facilities. Although Alzheimer's disease can occur at all stages of adulthood, most patients are in their sixties or older; only a small percentage are in their fifties or even younger...) |

Table 7: An example of our final SFT data.

## E  Pre-training Instruction: Knowledge Provision

As shown in Section 3.1, we use external LLMs (like ChatGPT) for data unification for two main purposes:

1. **Data Unification using ChatGPT:** The primary data source at this stage is raw medical corpora, not ChatGPT. ChatGPT is merely utilized for data refinement and back translation. Its function is similar to rule-based data polishing software or a fine-tuned T5 model, where medical expertise is not essential. We will subsequently present a case study using 'Baichuan2-Chat-7B' as an alternative for data unification.

2. **Fine-tuning Datasets using GPT-4 and ChatGPT:** Our fine-tuning data and ShareGPT are used to train our model to follow instructions and engage in conversation. However, they don't significantly contribute to the medical knowledge of our model. This aligns with the 'Superficial Alignment Hypothesis', suggesting that a model's knowledge and capabilities are primarily derived during its pre-training phase. Instruction fine-tuning mainly guides the model to apply specific formats and styles during user interaction, rather than substantially enhancing its knowledge base.

To illustrate, we conduct experiments to deeply analyze how using external large models affects HautuoGPT-II's medical knowledge. We sample 1/40 of our pre-training and fine-tuning data to assess their impact on the model's medical capability:

| Exper. No. | # Fine-Tuning Data | # Pre-Training Data | 2023 Exam (Pharmacist) | 2023 Exam (TCM) |
|:---:|:---:|:---:|:---:|:---:|
| #1 | 3.6K | 131.3K | 38.9 | 37.7 |
| #2 | 3.6K + **60K** | 131.3K | 37.7 | 38.1 |
| #3 | 3.6K | 131.3K + **60K** | 37.9 | 40.0 |
| #4 | 3.6K + **120K** | 131.3K | 37.3 | 35.7 |
| #5 | 3.6K | 131.3K + **120K** | **39.6** | **40.2** |

Table 8: Experiment on data scale.

The comparison between Experiments #4 and #5 shows that augmenting pre-training data is more effective than adding an equivalent amount of instruction tuning data. This suggests that the enhancement in medical knowledge benchmarks is primarily due to the raw data itself. ChatGPT play a role in transforming this raw data into formats that are more digestible for the trained LLMs, leading to our model outperforming ChatGPT and GPT-4 in medical benchmarks.

Furthermore, as outlined in Appendix M, leading Chinese medical models like DISC-MedLLM also employ ChatGPT and GPT-4 for their training data construction, yet our model surpasses them in domain benchmarks.

## F   Data Unification without External LLMs

To eliminate the performance improvement of our method caused by relying on external large language models, we experimente with adapting Baichuan2-7B-Chat in the medical domain using our method, without the assistance of other LLMs. We randomly select 5% of the pre-training corpus and medical queries, using Baichuan2-7B-Chat itself for data unification and direct response to the SFT-data's queries as fine-tuning data, a process we term as self-data-unification. The experiment is carried out as shown in Table F.

The results show that when using self-data-unification to perform self-data-unification, our method named One-stage Adaptation still exceeds Two-stage Adaptation by an average of 6.6 points. This

demonstrates our method's ability to function independently of other LLMs, effectively enhancing the model's domain adaptation capabilities and outperforming traditional continuous pre-training.

| | CMExam | 2023 Exam (Pharmacist) | 2023 Exam (TCM) |
|---|---|---|---|
| Two-stage Adaptation on Baichuan2-Chat-7B (Without Data-Unification) | 40.5 | 33.5 | 34.8 |
| Our One-stage Adaptation on Baichuan2-Chat-7B (With Data-Unification) | 48.3 | 40.0 | 40.3 |

Table 9: Adaptation gaps between Two-stage and On-stage adaptation.

Moreover, we find even Baichuan2-7B-Chat demonstrates a good ability to perform data-unification, as shown in the Table 10:

| Description | content |
|---|---|
| Raw Data（from PubMed) | The functional and structural alterations of the striatum in chronic spontaneous urticaria. The brain has long been known to be the regulation center of itch, but the neuropathology of chronic itch, su ch as chronic spontaneous urticaria (CSU)... |
| Query of Data Unification (Generated by Baichuan2-7B-Chat) | 在慢性自发性荨麻疹（CSU）中是否存在与临床症状相关的脑区活性和结构改变？（Translation: Are there alterations in the activity and structure of brain regions associated with clinical symptoms in chronic spontaneous urticaria (CSU)?) |
| Answer of Data Unification (Generated by Baichuan2-7B-Chat) | 在慢性自发性荨麻疹（CSU）中，确实存在与临床症状相关的脑区活性和结构改变。研究表明，大脑特别是纹状体（striatum）在调节瘙痒方面起着关键作用。对于慢性瘙痒，如慢性自发性荨麻疹（CSU）... (Translation: In chronic spontaneous urticaria (CSU), alterations in the activity and structure of brain regions associated with clinical symptoms do exist. Studies have shown that the brain, especially the striatum (striatum), plays a key role in the regulation of itch. For chronic itch, such as chronic spontaneous urticaria (CSU)...) |

Table 10: An example of QA data generated by Baichuan2-7B-Chat.

## G Ablation Study of Data Unification

| Setting | CMExam | 2023 Exam (Pharmacist) | 2023 Exam (TCM) |
|---|---|---|---|
| Two-stage, Without Data Unification | 50.5 | 38.8 | 38.3 |
| One-stage, Without Data Unification | 49.3 | 37.2 | 35.6 |
| One-stage, With Data Unification | 53.4 | 39.7 | 40.2 |

Table 11: Performance comparison: with vs. without Data Unification.

To validate the importance of Data Unification, we conducted an ablation study on Data Unification using a reduced dataset (5% due to time constraints) as shown in figure G.

The results emphasize the importance of Data Unification. Without it, the single-stage method falls short of the two-stage one. This could be due to the considerable discrepancies between the two data stages. Simply merging them doesn't yield effective results.

## H    Deviate Detection

In the data unification phase, we instruct LLM to refer to the text content to provide a detailed response. The responses are expected to mainly contain information from the text. However, the ability of the language models to follow instructions can't be fully guaranteed, and there might be instances where the model answers a question without referring to the text. To ensure that the response contains text knowledge, we adopt the following two methods to detect deviations from the original text:

1. **Statistical Method:** We convert both the text and the response into sets of 1-grams, and then use the Jaccard Similarity Coefficient to calculate the similarity between these two sets to determine the similarity of content between the text and the response. We set a threshold to detect content deviations.

2. **Model Detection Method:** The first method is suitable for detection within the same language, but it cannot handle cases where the English text is translated into a Chinese response. Therefore, we rely on a more robust method. We have people annotate whether an answer deviates and use this data to train a large language model for response content detection.

Based on these detection methods, if a generated response fails the tests, we instruct the Large Language Model (LLM) to regenerate it.

## I    Priority Sampling Strategy

In this section, we first explain how we handle ordering between pre-training and SFT data, and then give details of ordering within pre-training data

**Order between pre-training and SFT data**    We adopt a sequential training approach, prioritizing Pre-training Data followed by Fine-tuning Data (Pre-training Data→Fine-tuning Data). This method mirrors human learning processes: akin to initially learning from textbooks before attempting exercises. The rationale is straightforward  comprehensive foundational knowledge, gained from pre-training data, should precede the more specific applications found in fine-tuning data. This sequence prevents a scenario akin to attempting exercises without sufficient textbook study, which could lead to random guesses and, in the context of our model, the propensity to generate inaccurate or 'hallucinated' information. Additionally, the fine-tuning data, often structured in a question-and-answer format, closely aligns with real-world user interactions. By introducing this type of data later in the training process, we ensure that the model is better attuned to actual user scenarios, enhancing its practical applicability and effectiveness.

**Order within pre-training data**    Data quality is gauged using the data sampling epochs from the LLaMA (Touvron et al., 2023) study, as is shown in Table 12.

Initially, the significance of different data sources, as determined by sampling epochs in LLaMA, was ranked as "Web Corpus→Literature→Books→Encyclopedia". Nevertheless, after consulting with medical experts, it became clear that Books should be accorded greater importance. Consequently, we have adjusted our training sequence to: Web Corpus→Literature→Encyclopedia→Books. Lastly, we set the relative priority, a hyper-parameter in priority sampling.

| Dataset | Data Type | LLaMA Sampling Epochs |
|---------|-----------|----------------------|
| C4 | Web Corpus | 1.06 |
| ArXiv | Literature | 1.06 |
| Books | Book | 2.23 |
| Wikipedia | Encyclopedia | 2.45 |

Table 12: Sampling Epochs in LLaMA (Touvron et al., 2023).

## J Training Detail

For HuatuoGPT-II, $\beta$ was set to 2. In other settings, we set the sampling epoch for Medical Fine-tuning Instruction to 1 and for Medical Pre-training Instruction to 3. Additionally, we unified the data encoding. Sampled instructions were concatenated as much as possible to form a fixed-length token sequence of 4096 for training. The training was conducted with a batch size of 128 and a learning rate of 1e-4. As all the data are instructional, we only optimize the output loss and do not learn from the input loss. Our model is implemented in PyTorch using the Accelerate and leverage ZeRO algorithm to distribute the model. Training was conducted using 8 Nvidia A100 cards, and the HuatuoGPT-II (7B) training time was about 4 days.

## K Benchmark Details

We evaluate the medical capabilities of HuatuoGPT-II on popular benchmarks. Here, we select four medical benchmarks and three general benchmarks, noting that we only evaluate the medical part of the general benchmarks. The medical benchmarks include: MedQA (Jin et al., 2021), which is collected from professional medical board exams in various languages. We used its English test set for evaluation. MedMCQA (Pal et al., 2022), amassed from Indian medical entrance exams, and we evaluate it using the development set. CMB (Wang et al., 2023d), a comprehensive medical benchmark in Chinese, where we specifically used the CMB-Exam for evaluation. CMExam (Liu et al., 2023), a comprehensive Chinese medical exam dataset. The general benchmarks include: C-Eval (Huang et al., 2023), an all-encompassing Chinese evaluation framework. CMMLU (Li et al., 2023b), designed to critically appraise the knowledge and reasoning prowess of large language models in Chinese.

For the general benchmarks, we only use their medically related evaluation content. For CMMLU, the evaluation sections used are 'clinical knowledge', 'agronomy', 'college medicine', 'genetics', 'nutrition', 'Traditional Chinese Medicine', and 'virology'. For C-Eval, we use the 'clinical medicine' and 'basic medicine' parts.

Since all evaluations in our experiments are for Chat models, we uniformly adopted a Zero-shot setting, using a consistent form, shown as below:

---

```
请回答下面选择题。
对评估肝硬化患者预后意义不大的是
A．腹水
B．清蛋白
C．血电解质
D．凝血酶原时间
```

**Translation:**

Please answer the following multiple choice questions.

Of little significance in assessing the prognosis of a patient with cirrhosis is

A. ascites

B. albumin

C. blood electrolytes

D. prothrombin time

## L   Details of HuatuoEval

We utilize the single-round and multi-round evaluation data from HuatuoGPT (Wang et al., 2023b) to evaluate the model's medical response capability. Sources from these datasets include KUAKE-QIC (Zhang et al., 2021) for single-round questions and Med-dialog (Zeng et al., 2020) for multi-round cases. We slightly modify these evaluations for fairer assessment, focusing on the information in the answers rather than the tone of a doctor's response.

This evaluation is designed to evaluate the response capabilities of large-scale language models in medical scenarios. It includes two types of evaluations. The first type assesses the single-round answer capability, primarily comprising real patient questions sourced from KUAKE-QIC. The second type is a multi-round diagnostic evaluation, with data containing real patient case information, sourced from Med-dialog. In the single-round evaluation, we have the models directly answer medical questions. In the multi-round evaluation, we simulate a patient asking questions to a doctor using ChatGPT, based on the patient's medical record information. The simulated patient's prompt is shown as below( [Patient Case Information]  is patient case in multi-round data of HuatuoEval.). The models then engage in dialogue with the simulated patient to generate conversations. For a fair comparison, we have the ChatGPT-simulated patient continuously asking questions, and each model must respond twice.

| HuatuoGPT-II(7B) vs Other Model | Single-round QA | | | Multi-round Dialogue | | | Average Win/Tie Rate |
|---|---|---|---|---|---|---|---|
| | Win | Tie | Fail | Win | Tie | Fail | |
| HuatuoGPT-II(7B) vs HuatuoGPT-II(13B) | **39** | 22 | 39 | 41 | 13 | **46** | 57.5% |
| HuatuoGPT-II(7B) vs ERNIE Bot | **62** | 13 | 26 | **64** | 12 | 24 | 75.0% |
| HuatuoGPT-II(7B) vs ChatGLM3-6B | **75** | 11 | 14 | **76** | 10 | 14 | 86.0% |
| HuatuoGPT-II(7B) vs Baichuan2-7B-Chat | **75** | 7 | 18 | **84** | 7 | 9 | 86.5% |
| HuatuoGPT-II(7B) vs DISC-MedLLM | **80** | 8 | 12 | **73** | 15 | 12 | 88.0% |
| HuatuoGPT-II(7B) vs Qwen-14B-Chat | **82** | 7 | 6 | **79** | 9 | 12 | 91.0% |
| HuatuoGPT-II(7B) vs Qwen-7B-Chat | **89** | 6 | 5 | **75** | 12 | 13 | 91.0% |

Table 13: Results of the Automated Evaluation Using **GPT-4** in Chinese medical scenarios.

你是一名患者，下面是你的病情，你正在向医生咨询病情相关的问题，注意这是一个多轮问诊过程，切记不要让对话结束，要继续追问医生病情有关的问题。

[Patient Case Information]

**Translation:**

You are a patient, here is your condition and you are asking the doctor questions related to your condition, note that this is a multi-round questioning process, remember not to let the dialogue end, but continue to ask the doctor questions related to your condition.

[Patient Case Information]

## L.1 Automatic Evaluation

During automatic evaluation, we compare model responses in pairs. We present the dialogue or Q&A content of two models to GPT-4, which then judges which model's response is better. The prompts for single-round and multi-round automatic evaluations are shown in Table 14. To mitigate potential position bias in GPT-4 as a judge, each data point is evaluated for interaction position twice.

---

**The Prompt for Single-Round Automatic Evaluation:**

[Question]
[Question]
[End of Question]

[Assistant 1]
[The Response of Model 1]
[End of Assistant 1]

[Assistant 2]
[The Response of Model 2]
[End of Assistant 2]

[System]
We would like to request your feedback on the two AI assistants in response to the user question displayed above.
Requirements: The response should be to the point and adress the problem of user. The description of symptoms should be comprehensive and accurate, and the provided diagnosis should be the most reasonable inference based on all relevant factors and possibilities. The treatment recommendations should be effective and reliable, taking into account the severity or stages of the illness. The prescriptions should be effective and reliable, considering indications, contraindications, and dosages.
Please compare the performance of their responses. You should tell me whether Assistant 1 is 'better than', 'worse than', or 'equal to' Assistant 2.
Please first compare their responses and analyze which one is more in line with the given requirements.
In the last line, please output a single line containing only a single label selecting from 'Assistant 1 is better than Assistant 2', 'Assistant 1 is worse than Assistant 2', and 'Assistant 1 is equal to Assistant 2'.

---

**The Prompt for Multi-Round Automatic Evaluation:**

[Assistant 1]

[The Conversation from Model 1]

[End of Assistant 1]

[Assistant 2]

[The Conversation from Model 2]

[End of Assistant 2]

[System]
We would like to request your feedback on two multi-turn conversations between the AI assistant and the user displayed above.
Requirements: The response should be to the point and adress the problem of user. The description of symptoms should be comprehensive and accurate, and the provided diagnosis should be the most reasonable inference based on all relevant factors and possibilities. The treatment recommendations should be effective and reliable, taking into account the severity or stages of the illness. The prescriptions should be effective and reliable, considering indications, contraindications, and dosages.
Please compare the performance of the AI assistant in each conversation. You should tell me whether Assistant 1 is 'better than', 'worse than', or 'equal to' Assistant 2.
Please first compare their responses and analyze which one is more in line with the given requirements.
In the last line, please output a single line containing only a single label selecting from 'Assistant 1 is better than Assistant 2', 'Assistant 1 is worse than Assistant 2', and 'Assistant 1 is equal to Assistant 2'.

Table 14: The prompt for automatic evaluation on single-round and multi-round setting. Note that [.] is what needs to be filled in. 'Model 1' and 'Model 2' Indicates two models to be compared.

## L.2 Expert Evaluation

For manual evaluation, we provide licensed medical doctors with evaluation criteria shown as below. We then offer a platform for experts to conduct evaluations. Experts choose which response (from a pair of model responses) is better. The selection interface is shown in Figure 10. All model information is anonymized and interaction positions are randomized to ensure fairness.

1. 回复应该面向用户问题, 提供解决方案。
2. 考虑模型回复的丰富度, 逻辑清晰度。
3. 考虑模型的专业性, 准确性。
4. 模型回复应该富有人文关怀。

**Translation:**

1. The response should be orientated towards the user's problem and provide a solution.

2. consider the richness and logical clarity of the model response.

3. consider the professionalism and accuracy of the model.

4. the model response should be humanistic.

## M Baselines

**Open-Source Baselines** We compare to the most representative general large language models, which possess excellent chat capabilities and are adaptable to various scenarios, including healthcare. They are as follows:

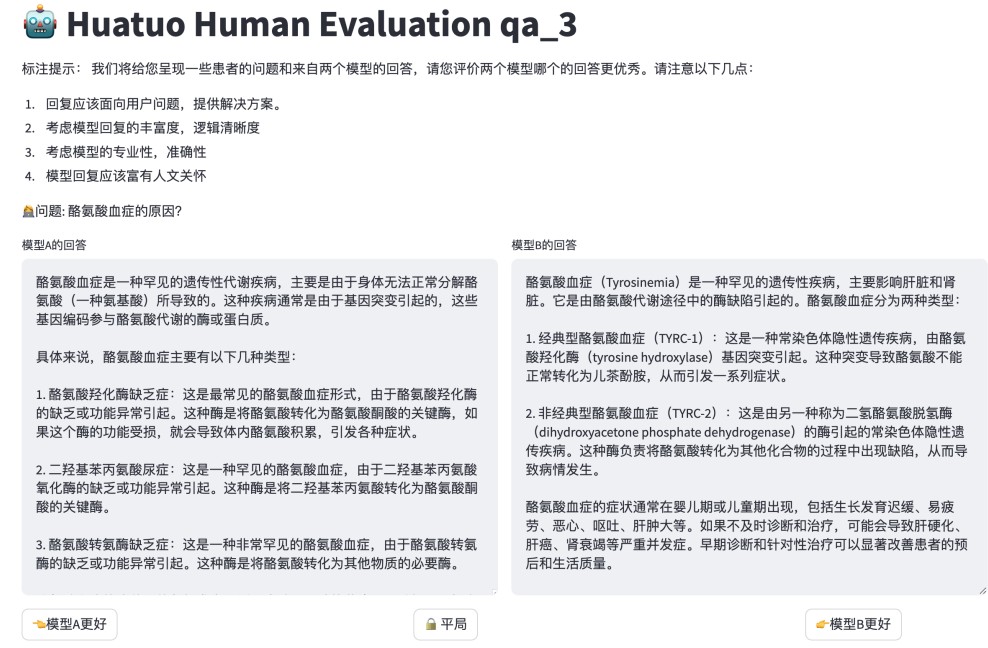

Figure 10: The interface of the expert evaluation.

- **Baichuan2-7B/13B-Chat**(Yang et al., 2023a) The Baichuan2-7B and Baichuan2-13B models are trained on 2.6 trillion tokens. Sharing the same backbone as ours, their chat-version models are well adapted to the base, fully leveraging its capabilities.

- **Qwen-7B/14B-Chat**(Bai et al., 2023) The Qwen series are trained on 3 trillion tokens of diverse texts and codes. The chat models are fine-tuned carefully on a dataset related to different tasks. RLHF is also applied to generate human-preferred responses.

- **ChatGLM3-6B**(Zeng et al., 2023) ChatGLM2-6B is a model trained with 1.4 trillion bilingual tokens. Based on ChatGLM2, the ChatGLM3 model has a more diverse training dataset, increased training steps, and a more reasonable training strategy.

- **Llama2-7B/13B-Chat** (Touvron et al., 2023) Llama2 series are successors models of Llama trained on 2 trillion tokens with diverse datasets. Their chat model has robust performance. Unlike the first three Chinese-supported models, Llama2 series are English models.

Additionally, we select two representative and strong large language models for Chinese medical scenario. (we conducted an experimental experiment to select these two models, see Appendix R for details). These models are:

- **DISC-MedLLM**(Bao et al., 2023) DISC-MedLLM is fine-tuned over 470K medical data and 34k general domain conversation and instruction samples.

- **HuatuoGPT**(Zhang et al., 2023) HuatuoGPT is trained using real-world data and distilled data from ChatGPT, adopting RLMF (a method combining ChatGPT and doctor preferences) to leverage the advantages of mixed data.

**Proprietary Baselines** Furthermore, we compare three representative proprietary models, which have larger parameters and stronger performance:

- **ERNIE Bot** ( 文心一言) (Sun et al., 2021) ERNIE bot is a closed-source large predictive model developed by Baidu. It is one of the strongest Chinese language models to date, with an API interface and a web version available for use.

- **ChatGPT**(OpenAI, 2022) ChatGPT is a large language model released by OpenAI, possessing significant influence and currently holding an excellent standard among large models.

- **GPT-4**(OpenAI, 2023) GPT-4 is a language model published by OpenAI following ChatGPT. It is undoubtedly one of the most powerful and widely used large language models currently available.

## N   Distribution of Synthesized Questions

We sampled 100 questions from the pre-training instructions and manually categorized their type as follows:

|  | Yes/No Question | Open-ended Question | Comparison Question | Causal Question |
|---|---|---|---|---|
| # Question | 12 | 47 | 9 | 32 |

Table 15:  Distribution of synthesized questions.

## O   Performance on Different Sizes of Models

The effect differences of HuatuoGPT2 with different model parameters:

| Model Size | Backbone | CMExam | 2023 Pharmacist | 2023 TCM |
|---|---|---|---|---|
| 7B | Baichuan2-7B | 65.8 | 47.7 | 47.5 |
| 13B | Baichuan2-13B | 69.0 | 52.9 | 51.6 |
| 34B | Yi-34B | 77.2 | 65.5 | 62.4 |

Table 16:  Performance on different sizes of models.

## P   Other SFT Data

We conducted experiments with different LLMs to generate SFT data, using 5% of the data, to verify the importance of SFT data. The results are shown in the table.

It shows minimal impact with the other LLM, which indicates that SFT data does not provide primary knowledge.

| | CMExam | 2023 Pharmacist | 2023 TCM |
|---|---|---|---|
| One Stage, with SFT data generated by GPT-4 | 53.4 | 39.7 | 40.2 |
| One Stage, with SFT data generated by ChatGPT | 52.6 | 40.0 | 39.8 |

Table 17: Results on different SFT data

## Q One Stage Training for Other Domains

The proposed one-stage training protocol is applicable beyond the Chinese medical domain.

To address the reviewer's question, we tested it on the MedQA-English (Jin et al., 2021) and CaseHOLD (Zheng et al., 2021) datasets, which represent the English medical and law domains, respectively. For MedQA-English, we used MedQA's English textbooks as the pre-training corpus and its training set for Supervised Fine-Tuning (SFT). For CaseHOLD, we sampled 4% of the corpora from FreeLaw Opinions (Biderman et al., 2022) as the pre-training corpus and used its training set for SFT. The data characteristics are as follows:

| Dataset | # Pre-training documents | # SFT data | # Test data | Domain |
|---|---|---|---|---|
| MedQA-en | 156,960 | 10,178 | 1,273 | English Medical Domain |
| CaseHOLD | 141,342 | 45,000 | 3,900 | Law Domain |

Table 18: Dataset characteristics in other domains.

We trained on Baichuan2-7B-Base and compared one-stage training with traditional two-stage training:

| Method | MedQA-en (English Medical Domain) | CaseHOLD (Law Domain) |
|---|---|---|
| Two-stage training | 46.2 | 41.8 |
| One-stage training | 48.6 | 43.6 |

Table 19: Performance comparison of training methods across domains

The results indicate that one-stage training can enhance performance across domains.

## R Other Medical Models

When selecting baselines for Chinese medical applications, we also tested the performance of other Chinese medical large models in the Chinese National Medical Licensing Examination. The results, as shown in Table 20, indicate that based on their superior performance, HuatuoGPT and DISC-MedLLM were chosen as the baselines for comparison.

## S Case Study

In our study, we observed that many models, including GPT-4, experience significant hallucinations in Chinese medical contexts. These hallucinations arise from two factors: 1) The model itself lacks specific medical knowledge; 2) Misconceptions arise in Chinese. Tables 21 and 22 present two examples of simple Chinese medical questions about pharmaceuticals.

| Model | Traditional Chinese Medicine (中医) | | | | | | | | | Clinical (临床) | | | | | | Avg. |
|---|---|---|---|---|---|---|---|---|---|---|---|---|---|---|---|---|
| | Assistant (执业助理医师) | | | Physician (执业医师) | | | Pharmacist (药师) | | | Assistant (执业助理医师) | | | Physician (执业医师) | | | |
| | 2015 | 2016 | 2017 | 2012 | 2013 | 2016 | 2017 | 2018 | 2019 | 2018 | 2019 | 2020 | 2018 | 2019 | 2020 | |
| DoctorGLM Xiong et al. (2023) | 3.0 | 1.4 | 3.5 | 1.8 | 1.8 | 2.1 | 1.8 | 2.4 | 2.2 | 2.6 | 3.3 | 1.6 | 1.6 | 2.7 | 1.8 | 2.2 |
| BianQue-2 (Chen et al., 2023a) | 3.7 | 2.3 | 3.5 | 4.2 | 4.5 | 4.0 | 7.1 | 4.2 | 9.0 | 4.7 | 5.3 | 1.6 | 3.8 | 5.9 | 3.7 | 4.5 |
| ChatMed-Consult (Wang et al., 2023c) | 20.0 | 17.3 | 14.7 | 21.3 | 18.2 | 20.0 | 16.7 | 15.0 | 17.5 | 27.8 | 21.3 | 19.3 | 23.8 | 21.4 | 19.5 | 20.0 |
| BenTsao (Wang et al., 2023b) | 23.3 | 26.8 | 17.2 | 19.0 | 19.5 | 22.1 | 20.2 | 20.4 | 18.8 | 18.4 | 24.6 | 27.5 | 21.6 | 18.9 | 18.1 | 21.1 |
| ChatGLM-Med (Wang et al., 2023c) | 20.7 | 23.2 | 20.7 | 21.8 | 21.8 | 22.6 | 16.7 | 21.0 | 15.3 | 30.3 | 23.0 | 29.9 | 18.5 | 20.4 | 24.8 | 22.0 |
| MedicalGPT (Xu, 2023) | 25.0 | 24.1 | 21.6 | 26.3 | 27.0 | 27.0 | 22.6 | 21.0 | 20.6 | 38.9 | 29.1 | 28.3 | 33.4 | 32.1 | 26.2 | 26.9 |
| HuatuoGPT (**Selected**) | 26.0 | 30.9 | 32.8 | 31.3 | 26.8 | 30.2 | 18.4 | 27.5 | 25.1 | 32.5 | 33.6 | 27.5 | 32.7 | 28.6 | 30.1 | 28.9 |
| DISC-MedLLM (**Selected**) | 38.3 | 41.8 | 26.7 | 36.2 | 38.7 | 35.1 | 25.0 | 22.2 | 22.0 | 49.2 | 36.1 | 41.8 | 41.4 | 36.6 | 35.8 | 35.1 |
| **HuatuoGPT-II (7B)** | 67.1 | 65.2 | 67.5 | 67.9 | 67.4 | 64.9 | 53.0 | 46.7 | 51.0 | 70.9 | 73.2 | 69.4 | 68.8 | 66.5 | 67.7 | 64.5 |

Table 20: The results of the Chinese National Medical Licensing Examination.

As shown in Table 21, GPT-4 seems to misunderstand the drug compound 'Methoxyphenamine', essentially providing irrelevant responses. Baichuan2-7B-Chat appears to comprehend this drug but only answers correctly in part, also exhibiting significant hallucinations. In contrast, HuatuoGPT-II accurately and comprehensively addresses the drug's details.

---

**Query:**

介绍下复方甲氧那明

( **Translation:** Introduce compound methoxyphenamine)

---

**GPT-4:**

复方甲氧那明是一种用于缓解流感和感冒症状的药物。它通常包含以下几种成分：
1. 甲氧那明：这是一种抗组胺药，用于减轻过敏症状，如打喷嚏、流涕和眼睛发痒。在复方甲氧那明中，它有助于缓解由感冒或流感引起的鼻塞和其他相关症状。
2. 对乙酰氨基酚（扑热息痛）：这是一种常见的镇痛剂和退热药，用于缓解疼痛和降低由感冒或流感引起的发热...

( **Translation:** Compound Methoxyphenamine is a medicine used to relieve flu and cold symptoms. It usually contains the following ingredients:
1. Methoxyphenamine: This is an antihistamine used to relieve allergy symptoms such as sneezing, runny nose and itchy eyes. In compounded methenamine, it helps to relieve nasal congestion and other related symptoms caused by a cold or flu.
2. Acetaminophen (Paracetamol): This is a common analgesic and antipyretic used to relieve pain and reduce fever caused by colds or flu... )

---

**Baichuan2-7B-Chat:**

复方甲氧那明是一种处方药，主要用于治疗各种炎症性疾病，如支气管炎、哮喘、过敏性鼻炎等。它是由四种成分组成的复方制剂：甲氧那明（一种支气管扩张剂）、氯苯那敏（一种抗组胺药）、氢氧化铝（一种胃酸抑制剂）和甲基硫酸吡啶酸（一种非处方解热镇痛药）...

( **Translation:** Compound Methoxyphenamine is a prescription medication used to treat a variety of inflammatory diseases, such as bronchitis, asthma, and allergic rhinitis. It is a combination of four ingredients: Methoxyphenamine (a bronchodilator), chlorpheniramine (an antihistamine), aluminum hydroxide (a gastric acid inhibitor), and pyridinium methylsulfate (an over-the-counter antipyretic and analgesic)...)

---

**HuatuoGPT-II(7B):**

复方甲氧那明是一种复方制剂，由盐酸甲氧那明、那可丁、氨茶碱和马来酸氯苯那敏那敏组成。它主要用于治疗支气管哮喘和喘息性支气管炎等呼吸系统相关疾病。

在使用复方甲氧那明时，可能会出现一些不良反应。例如偶尔会出现皮疹、皮肤发红、癌痒、恶心、呕吐、食欲不振、眩晕、心悸和排尿困难等。如果出现这些不口、良反应，应及时停药并咨询医生...

( **Translation:** Compound Methoxyphenamine is a compounded preparation consisting of methoxyphenamine hydrochloride, noscapine, aminophylline and chlorpheniramine maleate. It is mainly used for the treatment of respiratory related diseases such as bronchial asthma and wheezing bronchitis.

Some adverse reactions may occur when using Compound Methocarbamol. Examples occasionally include rash, redness of the skin, itchy cancer, nausea, vomiting, loss of appetite, dizziness, palpitations and difficulty in urination. If these not mouth, good reactions occur, should stop the drug and consult a doctor...)

Table 21: Chinese medical hallucinations case. The green font indicates
correct information, and the brown font indicates incorrect information.

Another example, as illustrated in Table 22, involves a direct question in Chinese about 'OxyContin'. GPT-4 erroneously associates it with 'Oscillococcinum', providing misleading information. Meanwhile, Baichuan2-7B-Chat experienced more severe hallucinations, referring to a non-existent drug named 'Oscilloclasm'. HuatuoGPT-II, however, correctly understood this as 'Oxycodone' and provided accurate information.

**Query:**

奥斯康定是什么药物

( **Translation:** What is the medication OxyContin?)

**GPT-4:**

奥斯康定（Oscillococcinum）是一种用于流感症状治疗的顺势疗法药物。它通常用于轻微或初期的流感症状，如发烧、寒战、疲倦和头痛。奥斯康定的主要成分是从鸭肝和鸭心提取的物质，这种提取物被高度稀释，以符合顺势疗法的原则。

( **Translation:** OxyContin (Oscillococcinum) is a homeopathic medicine used for the treatment of flu symptoms. It is usually used for mild or initial flu symptoms such as fever, chills, fatigue and headache. The main ingredient in Oscillococcinum is a substance extracted from the liver and heart of ducks, and this extract is highly diluted to comply with homeopathic principles.... )

**Baichuan2-7B-Chat:**

奥斯康定（Oscilloclasm）是一种抗癫痫药物，主要用于治疗各种类型的癫痫发作。它的化学名为"2-[(3,4-二甲氧基苯甲酰)氨基]-2-氧代丙酸"，其结构式为C16H17NO4，分子量为285.35。

(**Translation:** OxyContin (Oscilloclasm) is an antiepileptic drug that is primarily used to treat various types of seizures. Its chemical name is "2-[(3,4-dimethoxybenzoyl)amino]-2-oxopropionic acid", its structural formula is C16H17NO4, and its molecular weight is 285.35.)

**HuatuoGPT-II(7B):**

奥斯康定（Oxycodone）是一种强效的阿片类镇痛药，主要用于治疗中度到重度的疼痛。阿片类药物主要通过影响大脑中的阿片受体来减轻疼痛，这些受体主要存在于大脑和脊髓中。
奥斯康定在医疗领域有广泛的应用，包括用于术后疼痛、癌症疼痛、慢性疼痛等。然而，由于其强效的镇痛效果，奥斯康定也有可能被滥用，导致成瘾问题...

(**Translation:** OxyContin (Oxycodone) is a potent opioid analgesic used primarily to treat moderate to severe pain. Opioids reduce pain primarily by affecting opioid receptors in the brain, which are found mainly in the brain and spinal cord.
OxyContin has a wide range of applications in the medical field, including for post-operative pain, cancer pain, and chronic pain. However, due to its potent analgesic effects, OxyContin also has the potential to be abused, leading to addiction issues...)

Table 22: Chinese medical hallucinations case. The green font indicates correct information, and the brown font indicates incorrect information.

These instances highlight GPT-4's limitations in the Chinese medical domain. We believe there is a critical need to enhance domain-specific capabilities, especially for sensitive topics like healthcare, and HuatuoGPT-II appears to be more adept at this during its current developmental phase.

