# OpenReview forum: "HuatuoGPT-II, One-stage Training for Medical Adaption of LLMs"
_colmweb.org/COLM/2024/Conference — COLM_

### Official Review · Reviewer_F1mt · 2024-05-09

**Rating:** 6
**Confidence:** 4
**Ethics Flag:** 1

**Summary:**

This paper introduces a one-stage domain adaptation technique that combines diverse data from pre-training and supervised stages into a concise instruction-output pair format for efficient knowledge injection. The proposed mechanism, according to the authors, improves both training stability and domain generalization. Extensive experiments show the effectiveness of the method in four medical benchmarks (MedQA, MedMCQA, CMB, CMExam) and two general benchmarks (C-Eval, CMMLU).

**Questions To Authors:**

The authors claim that the proposed protocol improves training stability and domain generalization. What potential effects might it have on medical fields other than Chinese or other domains (e.g., law, finance and so on)?

**Reasons To Accept:**

- The suggested approach seems straightforward and might be used for other domains as well.
- Attempted to solve a critial issue of catastrophic forgetting in language model.
- The paper performed detailed experiments and ablation study to validate the effectiveness of the proposed method.

**Reasons To Reject:**

- The model, HuatuoGPT-II, specically developed only for the medical domain in Chinese. I have reservation with the claims of training stability and domain generalization. Please refer to 'Questions To Authors' for more details.
- The quality of the writing is a little sloppy. For instance, the overuse of "Camel Case," which is superfluous, detracted from the quality of writing. Furthermore, it is wrong to say '... let **an** Large Language Model (LLM) generate...' and unusual in academic writing to say '... and close to the data’s original content...'. I would like the authors to carefully review these minor issues throughout the manuscript.

---

> ### Author Rebuttal · Authors · 2024-05-31
>
> Thank you for your valuable comments. Please see our responses below.
>
> > Q1: The claim and the effects on other domains.
>
> ### Training Stability and Domain Generalization
> **Training stability** involves reducing data variability by transforming chaotic pre-training data into consistent QA format that aligns closely with SFT data. This results in a steadily decrease in training loss, as depicted in Figure 5.
>
> Regarding **domain generalization**, we realize our explanation was vague. We aimed to convey that one-stage training can better generalize domain-specific knowledge to downstream tasks. We will revise this statement to avoid confusion.
>
> ### Effects on Other Domains
>
> Our one-stage training protocol is applicable beyond the Chinese medical domain.
>
> To address the reviewer’s question, we test it on the *MedQA-English* (Jin et al., 2020)  and *CaseHOLD* (Zheng et al., 2021)  datasets, representing the English medical and law domains, respectively. For *MedQA-English*, we used *MedQA*'s English textbooks as the pre-training corpus and its training set for SFT. For *CaseHOLD*, we sample 4% corpora of *FreeLaw Opinions* (Biderman et al., 2022) as the pre-training corpus and used its training set for SFT. Data characteristics are as follows:
>
> |  | # Pre-training documents | # SFT data | # Test data | Domain                 |
> | - | -| - | - | -|
> | MedQA-en   | 156,960                  | 10,178     | 1,273       | English Medical Domain |
> | CaseHOLD   | 141,342                 | 45,000     | 3,900       | Law Domain |
>
> We trained on Baichuan2-7B-Base and compared one-stage training with traditional two-stage training:
>
> | Method         | MedQA-en (English Medical Domain) | CaseHOLD (Law Domain)
> | -- | -- | - |
> | Two-stage training (Continued Pre-training + SFT) | 46.2   |  41.8  |
> | One-stage training (Ours)      | 48.6              | 43.6 |
>
> The results indicate that one-stage training can enhance performance across domains.
>
> > Q2. Please address the issues in writing quality noted, such as the misuse of "Camel Case" and other linguistic inaccuracies.
>
> We acknowledge and appreciate your feedback on the writing quality of our manuscript, including the misuse of "Camel Case" and several grammatical errors. We commit to a thorough revision to address these issues and to ensure that our manuscript meets the high standards of academic writing. Your feedback is appreciated and will certainly help improve the quality of the paper.

---

> > ### Comment · Reviewer_F1mt · 2024-06-05
> > **Acknowledgment on rebuttal**
> >
> > Thanks for your efforts in preparing the rebuttal. I have read the authors' feedback and other reviews. I would prefer to maintain my initial assessment.

---

### Official Review · Reviewer_j9Xe · 2024-05-11

**Rating:** 6
**Confidence:** 4
**Ethics Flag:** 1

**Summary:**

This work presents a one-stage domain adaptation pipeline that relies on instruction tuning in place of the more common two-stage pipeline comprising continual pre-training and supervised finetuning. This approach aims to offer several advantages: a form of simplification, a means of reducing catastrophic forgetting, and mechanism to adaptively adjust data source during training. To test the efficacy of this model, there are numerous experiments (and a rich appendix) to substantiate claims.

**Questions To Authors:**

Suggestions:
- Carefully consider whether there are results that appear in the appendix that would be better included in the main text.
- This is stylistic, but consider removing some parentheticals where not necessary, e.g. "(necessitating dual meticulous training)".

Questions:
- What are future directions can be highlighted in discussion that are revealed by differences between Huatuo and Huatuo-II?
- If a different LLM is used in place of GPT-4 for generating instruction tuning data, how significantly does this impact the quality of the resulting model?
- How is performance affected with a very small (e.g. 1B) model or much larger model?

**Reasons To Accept:**

This work demonstrates several strengths:
- There are many experiments that aim to substantiate the claims (and a significant appendix) that helps piece together several parts of the story.
- The notion of simplification and adding a "knob" to more easily adjust data mixing are both quite useful in the context of model adaptation and could potentially help mitigate multiple distribution shifts using a consistent method.
- The use of other tools, e.g. GPT, to construct instructions introduce an additional source of information that can be scaled more easily than the traditional data generation process.

**Reasons To Reject:**

There are some considerations that limit impact, namely:
- Prior work (Peng et al., https://arxiv.org/pdf/2304.03277) demonstrated the efficacy of instruction tuning with GPT-4, which would be useful to contextualize the present work. Further, it would be useful to add discussion around whether this setup might suggest that GPT-4 offers an upper found on performance for these tasks were it similarly finetuned. On the one hand, the experimental results suggest HuatuoGPT-II has a superior win rate, but it's not quite an equal comparison.
- Adding clarity around the underlying distribution shifts that are present would be useful because there is both a significant shift in language and general vs medical domain present in this work. In particular, the evaluation chosen relies on resources like MedQA which are primarily clinical questions (from medical exams) whereas the training data is drawn from biomedical (as opposed to clinical) domain.
- Focus the discussion of the results to emphasize the key takeaways. Currently there is a rich appendix that substantiates several claims made throughout the text but the main paper is difficult to read in absence of the appendix because key details have been relegated there, some of which could arguably be considered more important than those that appear directly in the results, such as Appendix N (Other Medical Models), which provides context for what performance means in this space.

---

> ### Author Rebuttal · Authors · 2024-05-31
>
> We sincerely appreciate your valuable comments. Below, we provide responses to your concerns.
>
> > Q1: The role of GPT-4 and fairness.
>
> We appreciate your suggestions. We will expand the discussion as follows:
>
> **Upper Bound** GPT-4 is not the upper bound of our models in the Chinese medical domain, as they learn from domain-specific corpora rather than relying solely on GPT-4.  *Appendix E* shows that the SFT data (generated by GPT-4) is not essential for domain expertise.
>
> **Fairness of Comparison** We strive to enhance the fairness of evaluations by utilizing expert assessments and the latest exams.
>
>
> > Q2: Clarify the distribution shifts.
>
> Our training data is rich and varied, containing not only biomedical content but also clinical content, such as web corpora and textbooks. We value your suggestion and will clarify the distribution shifts from general to medical.
>
>
> > Q3: The suggestion about key appendices.
>
> Thank you for your advice and feedback! In the next version, we will endeavor to move important appendices to the forefront to enhance readability.
>
> > Q4: Future directions about HuatuoGPT.
>
> The evolution from Huatuo to Huatuo-II highlights our focus on two critical areas:
> - **Enhancing Trustworthiness**: Minimizing hallucinatory outputs is crucial in the medical domain. We are exploring retrieval-augmented generation and reliability scoring for trustworthiness.
> - **Enhancing Domain Capability**: We are advancing their abilities to analyze medical images, such as CT and MRI scans, and improve decision-making abilities in diagnostic support.
>
>
> > Q5: Other LLMs to generate SFT data.
>
> In response to your concern, we conducted experiments with different LLMs to generate SFT data, using 5% of the data due to time constraints. The results are as follows:
>
> || CMExam | 2023 Pharmacist | 2023 TCM |
> |-|-|-|-|
> | One Stage, with SFT data generated by *GPT-4* | 53.4| 39.7| 40.2|
> | One Stage, with SFT data generated by *ChatGPT* | 52.6 | 40.0 | 39.8 |
> It shows minimal impact with the other LLM (ChatGPT).
>
> > Q6: Performance on smaller or larger models.
>
> We will train a tiny HuatuoGPT-II to assess the smaller models. In fact, there is a 34B version of HuatuoGPT-II (based on Yi-34B), and its performance compared is as follows:
> | Model Size | CMExam | 2023 Pharmacist | 2023 TCM |
> |-|-|-|-|
> | 7B | 65.8 | 47.7 | 47.5 |
> | 13B | 69.0 | 52.9 | 51.6 |
> | 34B | 77.2 | 65.5 | 62.4 |
> The results demonstrate that larger model parameters can significantly enhance performance.

---

> > ### Comment · Reviewer_j9Xe · 2024-06-05
> > **Acknowledgement of Rebuttal**
> >
> > Thank you for answering these questions. In particular, the answers to Q5 and Q6 are quite interesting and I would encourage the inclusion of these results, whether in the work itself or an appendix if space doesn't permit. For Q2, please do try to add clarity around this point in revision but it's not necessary to do so here.

---

### Official Review · Reviewer_FV3D · 2024-05-13

**Rating:** 6
**Confidence:** 4
**Ethics Flag:** 1

**Summary:**

The authors tackle the problem of domain adaptation of large language models (LLMs) by proposing a unified, one-stage adaptation process in the place of the more common two-stage approach of continued pretraining and supervised fine tuning. Their approach transforms the domain-specific pretraining corpus into a form that more closely resembles the instruction tuning data used during fine tuning. The authors select a filtered dataset of over 5 million documents in the medical domain to demonstrate this transformation process and how this data is used to adapt a general-domain LLM to the medical domain. The resulting model and competing baseline models are graded against a recent, previously unseen Chinese National Pharmacist Licensure Examination, with the hope that the contents of this test are not present in some form in either the general or domain-specific training data used by the models. Both automatic and human-expert evaluations are performed. Results show the authors' models generally outperforming the baseline models on Chinese-language tasks, including commercial offerings such as GPT-4.

**Questions To Authors:**

- Can you define what is meant by 'vertical domains' in Section 1?
- A footnote for a definition of 'Philosophy of Parsimony' may be useful
- "unpopular languages" may be better represented as "lower-resourced languages" in Section 1
- Section 3.1, corpus is -> corpora are

**Reasons To Accept:**

The unified one-stage approach for tuning a LLM to a specific domain is interesting, especially where the pretraining data is transformed into instruction-tuning-like data via interactions with a separate LLM (the authors note there is no strong reliance on a single model in this transformation step.)

**Reasons To Reject:**

While there has been great care to note that the licensing exam has been chosen for novelty (and was published after the data collection cut-off date), there may questions present in earlier published study guides that were then incorporated as training or tuning data. It may be worth a quick examination on scores from previous editions of the licensing exams for comparison.

---

> ### Author Rebuttal · Authors · 2024-05-31
>
> Thank you for reviewing our paper. Below, we provide responses to your concerns.
>
> > Q1: Concern about the latest exams.
>
> Thank you for raising a valid point regarding the potential overlaps with prior materials. We will add the following clarification to our paper:
>
> - **Uniqueness of the Exam:** According to the [official exam website](http://www.cqlp.org), the questions for the National Pharmacist Licensing Exam are strictly designed and newly formulated. This significantly reduces the risk of data leakage and ensures greater fairness.
>
> - **Comparison with Previous Exams**
> It is quite worthwhile to examine the scores from previous exams! In *Appendix B*, we provide the results of the pharmacist exams from the past three years for comparison:
>
> |                    | 2017 Exam | 2018 Exam | 2019 Exam | 2023 Exam (Latest) |
> | ------------------ | --------- | --------- | --------- | ----------------- |
> | Baichuan2-13B-Chat | 54.2      | 38.9      | 48.4      | 43.0              |
> | GPT-4              | 41.1      | 43.7      | 48.0      | 48.1              |
> | ERNIE Bot          | 55.9      | 50.3      | 60.0      | 50.6              |
> | HuatuoGPT-II (13B) | 56.3      | 52.1      | 54.7      | 52.3              |
>
> Note that the exam questions from 2017-2019 are incomplete, with details available in *Appendix B*.
>
> Nonetheless, we observe that: **(i)** HuatuoGPT-II scores are relatively close to those of previous exams, and **(ii)** some models (such as ERNIE Bot and Baichuan2-13B-Chat) scored notably higher in previous years compared to the 2023 exam.
>
> > Q2: "Vertical Domains" & "Philosophy of Parsimony"
>
> *“Vertical domains”* refer to specialized fields such as medicine, law, and finance, where training focuses on deep, domain-specific knowledge  and disregards other unnecessary domains. This contrasts with horizontally trained models, which possess broader but shallower general knowledge.
>
> We appreciate the reviewer's suggestion and will add a footnote explaining this *"Philosophy of Parsimony"*.
>
> > Q3: Terminology and grammatical corrections
>
> Special thanks to the reviewer for contributing to the improvement of our manuscript. We commit to correct these linguistic errors and thoroughly review our paper to enhance its quality.

---

> > ### Comment · Reviewer_FV3D · 2024-06-05
> > **Acknowledgement of rebuttal**
> >
> > Thank you for providing additional information and clarifications for your work, in particular regarding performance on previous years exams. At this time, I will concur with the other reviewers and maintain the current review score.

---

### Official Review · Reviewer_sbnB · 2024-05-26

**Rating:** 6
**Confidence:** 3
**Ethics Flag:** 1

**Summary:**

This paper introduces a one-stage domain-adaptation protocol for language models and produces a medical domain language model HuatuoGPT-II. Different from a two-stage process (pre-training, supervised fine-tuning), the paper proposes a one-stage approach to merge these two stages. The key technique here is to unify different format of data into instruction tuning pairs, leveraged by auto-labeling ability of large models (i.e., GPT-4). The paper also proposes a priority sampling strategy for training. The models are shown to perform better compared to other models on different Chinese medical benchmarks.

**Questions To Authors:**

- What types of questions are generated on the pre-training corpus?
- The paper makes an assumption that training with instruction data (which is generated on pre-training corpus) has the same functioning of training on pre-training corpus. What is the intuition behind it?

**Reasons To Accept:**

- The paper proposes a simple training protocol, which can indeed help reduce the efforts needed in a traditional two-stage training.
- The paper is easy to understand.
- HuatuoGPT-II shows strong performance on different benchmarks
- The paper also introduces a new Chinese Pharmacist Licensure Examination to avoid the test data leakage concerns for existing benchmarks.

**Reasons To Reject:**

- The connection between training on pre-training corpora and training on instruction data generated from pre-training corpora is not clear to me. But it seems to be a core inspiration of the methodology. I think providing more and deep insights will be useful.
- The baseline approaches are a bit weak as they are mostly general-domain models.
- Given the paper uses GPT-4 to generate data, uses GPT-4 to evaluate results, and compares the proposed model with GPT-4, it will be tricky to interpret the results, especially when comparing to GPT-4.

---

> ### Author Rebuttal · Authors · 2024-05-31
>
> We sincerely appreciate your valuable comments. Below, we address your concerns.
>
> > Q1: The Connection between Pre-Training Corpora and Pre-Training Instructions
>
> Both learning approaches aim for autoregressive goals, yet the differences between them are both subtle and significant:
>
> - **Vanilla Pre-Training** focuses on learning the probability of a sequence $ p(y_i \mid y_0, \ldots, y_{i-1})$ , primarily targeting next-token prediction. This method enables the model to learn *correlations* between words, with the expectation that it can generalize to downstream scenarios to handle various types of questions.
>
> - In contrast, **HuatuoGPT-II** aims to learn $p(y_i \mid X, y_0, \ldots, y_{i-1}) $, integrating the input $X$. This method learns a direct mapping from $ X $ to $ Y $, establishing a more high-level relationship (e.g. *causal*)  rather than merely *correlations*, which aligns more closely with knowledge understanding and the SFT objective.
>
> **Validation Experiment** As shown in *Appendix G*, training in a QA format  outperforms vanilla pre-training:
>
> |   | CMExam | 2023 Pharmacist | 2023 TCM |
> |-|-|-|-|
> | Training on *pre-training corpora* | 49.3 |37.2|35.6|
> | Training on *pre-training instruction* |53.4|39.7|40.2|
>
> > Q2: The use of baselines.
>
> We extra conduct a comparison of **eight** medical baselines in *Appendix N* for a more comprehensive domain-specific analysis.
>
> > Q3: Usage of GPT-4 in data generation and evaluation.
>
> **GPT-4 for Evaluation**
> In addition to using GPT-4 for evaluation, we also employed medical expert evaluations and the latest medical exams to ensure fairness.
>
> **GPT-4 in Data Generation**
> GPT-4 is used to generate SFT data for instruction-following. Moreover, *Appendix E* demonstrates that SFT data is not essential for domain expertise but rather for pre-training instruction.
>
> > Q4: Types of questions generated.
>
> We sampled 100 questions from the pre-training instructions and manually categorized their type as follows:
>
> | | Yes/No Question| Open-ended Question| Comparison Question| Causal Question|
> |-|-|-|-|-|
> | # Question |12| 47 | 9 | 32 |
>
> > Q5: Intuition behind training with the pre-training instruction.
>
> As outlined in response to  `Q1`, our intuition is that pre-training instruction facilitates a more direct method to learn explicit causal relationships, as opposed to mere word correlations in the pre-training. We appreciate this valuable inquiry and will clarify this further in our revised paper.

---

> > ### Comment · Reviewer_sbnB · 2024-06-06
> >
> > Thanks for responding to my review. I will keep my score.

---

### Decision · Program_Chairs · 2024-07-10

**Decision:**

Accept

**Comment:**

The paper on HuatuoGPT-II introduces a novel one-stage domain adaptation protocol that combines pre-training and supervised fine-tuning into a unified instruction-output pair format. The reviewers are happy with the paper for its clarity and the strong performance of HuatuoGPT-II on various Chinese medical benchmarks; they also express some concerns about the clarity of certain methodological aspects and some concerns about using GPT-4 for data generation and evaluation. The paper was rated 6 (marginally above the acceptance threshold), and the authors are asked to provide additional clarifications and address the raised concerns.

Comments from PC: Please include the additional results from the rebuttal in the revision, as they significantly help clarify the questions raised by the reviewers.